# Diagnostic Stewardship as a Team Sport: Interdisciplinary Perspectives on Improved Implementation of Interventions and Effect Measurement

**DOI:** 10.3390/antibiotics11020250

**Published:** 2022-02-15

**Authors:** Kyle D. Hueth, Andrea M. Prinzi, Tristan T. Timbrook

**Affiliations:** 1BioMérieux, Salt Lake City, UT 84104, USA; kyle.hueth@biomerieux.com (K.D.H.); andrea.prinzi@biomerieux.com (A.M.P.); 2College of Pharmacy, University of Utah, Salt Lake City, UT 84112, USA

**Keywords:** antimicrobial stewardship, diagnostic stewardship, infections, rapid diagnostics

## Abstract

Diagnostic stewardship aims to deliver the right test to the right patient at the right time and is optimally combined with antimicrobial stewardship to allow for the right interpretation to translate into the right antimicrobial at the right time. Laboratorians, physicians, pharmacists, and other healthcare providers have an opportunity to improve the effectiveness of diagnostics through collaborative activities around pre-analytical and post-analytical periods of diagnostic testing. Additionally, special considerations should be given to measuring the effectiveness of diagnostics over time. Herein, we perform a narrative review of the literature on these potential optimization opportunities and the temporal factors that can yield changes in diagnostic effectiveness. Our objective is to inform on these considerations to ensure enhanced value through improved implementation and measurement of effectiveness for local stakeholder metrics and/or clinical outcomes research.

## 1. Introduction

Over the past decade, there has been an increased focus on improving the quality of care with the aim of improving both outcomes and reducing overall healthcare costs. In this pursuit, the importance of interdisciplinary care delivery has been highlighted. Diagnostic testing influences many patient care decisions and thus should be a central focus of efforts to improve outcomes and reduce the overall cost of care [1]. In 2015, the Institute of Medicine (IOM) published their report Improving Diagnosis in Health Care to highlight the importance of the diagnostic process [2]. Several goals are outlined in this report including improved interdisciplinary teamwork, employing processes to evaluate and reduce diagnostic errors, and the establishment of processes to ensure effective and timely communication between diagnostic testing, healthcare professionals, and treating clinicians. These goals highlight the importance of diagnostic stewardship.

Diagnostic stewardship encompasses activities that span the diagnostic process from pre-analytic to post-analytic with inputs from several disciplines including laboratory medicine, pharmacy, nursing, and treating clinicians and aims to ensure appropriate tests are ordered and resulting data are actionable and timely. In this narrative review we will focus on the perspectives from clinical microbiology and antimicrobial stewardship to describe important considerations in the pre-analytic phase, post-analytic phase, and stewardship effect measurement (Table 1).

## 2. Considerations for Pre-Analytical Diagnostic Stewardship Interventions

The selection of diagnostic testing is a critical initiator of the clinical management pathway and appropriate test selection has considerable influence on reaching the correct diagnosis and subsequent appropriate clinical decisions [6]. There are a myriad of factors ranging in complexity that influence test selection including test naming conventions, laboratory formularies, order sets, and lack of provider knowledge of test performance, and these must be thought through when implementing a new test. A fundamental activity that must be undertaken is providing education to the clinical teams before a test is brought online. This includes educating on the intended use of a test, appropriate specimen type and collection techniques, test performance characteristics (i.e., sensitivity, specificity), and also expected turnaround time for the result.

With much of healthcare focused on reducing waste, appropriate utilization of diagnostic testing tends to be the first thought in optimizing utilization. The American Board of Internal Medicine’s Choosing Wisely campaign, with input from the American Society for Clinical Pathology, focuses on appropriate test selection and the timing of the test [7]. Appropriate utilization is indeed an important focus as it results in improved costs, appropriate diagnosis, and avoidance of diagnostic uncertainty; however, underutilization and inappropriate utilization (i.e., inadvertent selection of the wrong test or in absence of appropriate syndrome or symptoms) should also be a focus for diagnostic stewardship teams [8]. Understanding the utilization challenges is key to successfully implementing a new diagnostic test. 

### 2.1. Optimizing Test Utilization

Appropriate utilization is testing that improves health outcomes or clinical decision-making and can lead to improved costs, appropriate diagnosis, and treatment [9]. There are several challenges associated with appropriate test utilization, and central to these is a test menu that is rapidly increasing in volume and complexity. These challenges are particularly relevant for molecular testing, where the menu has expanded the availability of singleplex assays, multiplex assays, and point-of-care tests, even for a single given pathogen. With several tests appearing to be applicable, it is easy to understand how an ordering provider may inadvertently order duplicate tests. To ensure appropriate utilization of newly implemented technologies, laboratories should routinely audit their test menus in consultation with the clinical team to evaluate the relevance of other tests that are offered to have a simplified and clinically relevant test menu. In addition, the inappropriate inclusion of a test within an order set is an important driver of overutilization. Diagnostic stewardship teams should be engaged in the inclusion of new technologies in order sets to avoid and reduce the potential for overuse which may limit the availability of the technology in the future. Finally, interventions to address duplicate orders may help avoid excess utilization. Laboratories should consider the drivers of appropriate utilization discussed here to implement a duplicate order canceling mechanism in the laboratory information system (LIS). For example, a study at an academic center implemented an intervention to block duplicate orders and demonstrated avoidance of 11,790 tests [10]. Furthermore, providers called to insist on the order in only 3% of cases demonstrating this was an effective intervention compatible with the computerized physician order entry (CPOE) workflow.

Underutilization, or under testing for a condition or pathogen, should also be an essential diagnostic stewardship consideration when implementing a new test. Delay in reaching a definitive diagnosis can result in inappropriate or unnecessary care, additional testing, and extended length of stay. Selecting the appropriate test can be particularly difficult when the clinical presentation overlaps several etiologies. Rapid molecular diagnostics, generally defined as providing a result within 4–6 h, that test for a range of potential pathogens are becoming increasingly ubiquitous and offer much promise for improving outcomes [11]. These panels testing for multiple pathogens with a single test have improved diagnostic yield compared to the standard of care where one to many individual tests are ordered. As an example, a multiplex PCR for meningitis/encephalitis has demonstrated improved diagnostic yield in several studies with a reduction in length of stay and improved therapy (i.e., discontinuation of empiric antimicrobials where appropriate) also demonstrated [12,13,14,15,16]. Similarly, for neurological infections without a diagnosis due to uncommon pathogens not identified by routine methods, metagenomic next-generation sequencing (NGS) has shown promise in identifying rare etiologies [17]. Stewardship teams should ensure the intended use of new technologies is understood by the care teams and appropriately represented in the ordering system to ensure appropriate utilization for maximum benefit to patients. 

### 2.2. Using LIS Data and Clinical Decision Support Interventions to Support Improved Test Utilization

Diagnostic testing utilization trends are influenced by various factors that differ significantly between and within institutions [18,19]. An important activity for the diagnostic stewardship team is to understand their institution’s utilization trends and practices, particularly when a new test is being offered. The laboratory produces a significant proportion of the data in the medical record, and LIS data are well suited to support this activity. Laboratorians should leverage these data to identify variations in utilization to compare providers, specialties, and institutions, and outliers should be further evaluated [20]. Using these data to provide feedback and targeted education to providers has demonstrated improved utilization, particularly in conjunction with other initiatives [21,22]. This is a straightforward strategy to begin identifying opportunities that can serve as the basis of conversations with clinical teams. Additionally, a literature review to identify recommendations and guidelines that can be used to benchmark trends can be very compelling. Successful literature review application has been demonstrated in evaluating repeat testing where appropriate retest intervals were derived from the literature and applied to utilization trends to determine compliance and identify potentially redundant tests [23]. 

Clinical decision support and configuration of the CPOE is perhaps the most crucial consideration before bringing on a new test to promote appropriate utilization. As discussed, the lab test menu is ever-expanding and complex. This poses a challenge for infectious disease testing where culture-based tests, antigen assays, serology, or molecular methods are available. Therefore, special consideration should be given to how the test is named and what information accompanies the orderable in the system. Test naming conventions have demonstrated significant appropriate utilization challenges [24,25,26]. Recently, the initiative Test Renaming for Understanding and Utilization for Laboratory Test Names (TRUU-Lab) has begun and represents a concerted effort with participation across industry and government agencies to address this challenge [27].

### 2.3. The Importance of Laboratory Requirements for Specimen Acceptability in the Pre-Analytical Phase

Minimum specimen requirements have long been an essential quality component of the pre-analytical phase in the microbiology laboratory and remain so in the face of emerging diagnostic technologies. For example, rejecting formed stools for *Clostridioides difficile (C. difficile)* PCR testing remains a key measure in ensuring that a positive *C. difficile* result represents clinical infection as accurately as possible [28]. Furthermore, studies have demonstrated that screening stool specimens for recent *C. difficile testing* and only accepting those from patients with new clinical symptoms or a *C. difficile* test greater than seven days prior reduces the number of unnecessary tests and unhelpful results [3,29,30]. Implementing minimum acceptability criteria for cerebrospinal fluid (CSF) that includes laboratory values (i.e., glucose, protein, white blood cells), patient immune status, and seasonality of disease may help prevent test overuse and guide results interpretation [31]. Finally, using the Gram stain to screen the quality of specimens from nonsterile sites, such as lower respiratory specimens, can help ensure that results from highly sensitive assays are more clinically relevant and interpretable. Failing to screen these specimens for quality prior to culture or PCR can lead to the inappropriate interpretation of results and excessive antimicrobial use directed toward commensal respiratory flora [32].

These pre-analytic considerations are crucial when implementing new technology and the prospective maintenance of test utilization. Failure to address these considerations may lead to suboptimal utilization, diluting the overall cost–benefit of the diagnostic technology, potentially limiting its availability in the future. This highlights the fundamental role of diagnostic stewardship.

## 3. Considerations for Post-Analytic Diagnostic Stewardship Interventions

Results reporting from the laboratory is arguably one of the most critical targets for stewardship interventions. The advent and implementation of rapid diagnostics in the microbiology laboratory have highlighted the importance of providing accurate, timely, and actionable results to the clinical team. While delivering such results is an essential component of effective stewardship programs, research has demonstrated that the impact on care or outcomes may be reduced if results are not paired with appropriate clinical follow-up [33,34]. The impact of antimicrobial stewardship team (ASP) involvement on rapid test results has been well-described for diagnosing bloodstream infections (BSI). In children, the implementation of rapid diagnostic technology for the diagnosis of BSIs combined with antimicrobial decision support is associated with a decreased time to therapy and optimization of antimicrobial use as well as high provider satisfaction ratings [35]. A 2017 meta-analysis of studies evaluating the impact of rapid diagnostics for BSIs demonstrated that their use was significantly associated with a decreased mortality risk when paired with an antimicrobial stewardship program (ASP), but not in the absence of one [34]. Additionally, rapid results delivery to the clinical team may contribute to better outcomes. For example, in a pre–post study of adults admitted to the intensive care unit (ICU) with a positive blood culture organism identified by rapid diagnostic assay, in-basket notifications of positive results were not significantly associated with improved antibiotic use or clinical outcomes, suggesting that active interventions and open communication with the microbiology laboratory are needed to optimize the impact of rapid diagnostic tests [36].

In the case of pneumonia, multifaceted stewardship interventions have successfully improved antimicrobial use. For example, a recent study by Moradi et al. examined the impact of an electronic health record-based best practice alert (BPA) tailored toward advising clinicians on appropriate antibiotic use according to patient’s procalcitonin and rapid respiratory panel results [37]. This single-center study demonstrated a 19.2% higher rate of antibiotic discontinuation in patients with respiratory illness within 24 h of the alert firing. In addition, the study proposed a minimally invasive intervention that can be utilized throughout the day, even when ASP team members are unavailable or not working. Even in the absence of rapid diagnostic technology, optimizing how microbiology results are communicated to clinicians can positively impact downstream outcomes. The microbiology laboratory is a core component of any stewardship program [38], and simple modifications to post-analytic communication of microbiology results can significantly influence provider behavior. For example, Musgrove et al. demonstrated that instead of only reporting “commensal respiratory flora” from respiratory cultures and specifying that the commensal flora did not include *Staphylococcus aureus* or *Pseudomonas aeruginosa*, the use of broad-spectrum antibiotics used to target those organisms was decreased [39]. The success of this intervention was further amplified using in-person and written education to ensure clinician adoption and understanding. Similar studies suggest that microbiology laboratory nudges and selective reporting effectively alter antimicrobial prescribing and optimize diagnostic test use [40,41]. Finally, for some diagnostics, evidence-based guidelines or protocols used at the point of care may help apply the microbiology results to clinical decision making and is therefore supported in the American Society for Clinical Pathology’s “Choosing Wisely” recommendations [42].

Ultimately, the antimicrobial stewardship literature tells us that implementing diagnostics alone may not be enough to influence behavior change or fully demonstrate the value of such assays. Interventions that involve actionable diagnostic testing combined with real-time clinical follow-up and communication with the microbiology laboratory are more likely to facilitate change. Often, an impactful intervention will require a variety of disciplines and tactics and should be modifiable to various contexts and practices.

### 3.1. Designing Implementations for Success and Sustainability: The Importance of Local Context

Building on the promise of the impact of clinician nudging, researchers have begun to focus their studies on the psychology and behavioral economics of antimicrobial prescribing practices [43,44]. While BPAs and educational interventions may modestly impact prescribing practices, they also have been associated with information overload. As a result, they may serve as barriers to a clinician’s workflow, diluting their potential positive impact [43]. These findings serve as a potent reminder that clinical settings are heterogeneous and that how clinicians and laboratorians respond to stewardship interventions will differ based on various factors.

Complex specimens such as respiratory samples help paint a picture of the challenges of implementing impactful quality improvement initiatives to reduce excessive test and antimicrobial use. Results from a survey of microbiology laboratories across the United States demonstrate marked variability between and among laboratories in every stage of the respiratory culture process [45]. Factors such as clinician preference, local epidemiology, and availability of laboratory resources drive this process variability and result reporting. Understanding the impact of each of these factors at the local level is critical to the success of any laboratory-based stewardship intervention. In clinical practice, a recent survey highlights the variability in knowledge and opinion among hospitals and between departments within the same hospital [46]. Despite most clinicians stating that they feel knowledgeable about testing and treating guidelines, significant differences in opinion remain regarding the use of rapid respiratory diagnostics. Ultimately, individual clinician confidence in decision making, familiarity with the diagnostic test, department, and the level of education clinicians receive about a test impacts the way a clinician interprets a rapid respiratory diagnostic result [46]. Without evaluating all the possible factors that could contribute to the success or failure of an intervention in the post-analytic phase of infectious disease diagnostics, the impact and sustainability of stewardship interventions are at risk of being less impactful.

### 3.2. The Future of Successful Stewardship: D&I Science and Laboratory Collaboration

Dissemination and Implementation Science (D&I) is a relatively new field of research that combines evidence-based interventions with elements of quality improvement. The overarching goal of the D&I field is to improve the quality and effectiveness of health services through the rapid uptake of research findings into routine practice [47]. While the D&I methodology has been widely applied to population health, its place in antimicrobial and diagnostic stewardship cannot be overlooked. Unlike randomized, controlled trials, which may not always reflect real-world situations, D&I research takes place at the third and fourth levels of translational science, where clinical interventions are evaluated in real time.

Many D&I frameworks support investigators in accounting for local context while studying the impact of clinical interventions, encouraging the successful development of stewardship strategies [47]. For example, critical stakeholders for a diagnostic stewardship intervention in the laboratory may include clinical end-users, microbiologists, administrators, and IT professionals. In contrast, stakeholders for a clinical intervention may include the laboratory, administrators, other clinicians, and patients. The use of recommended D&I frameworks for stewardship interventions may help identify the role of each of these stakeholders and prevent siloed research or quality improvement.

For example, the Practical, Robust, Implementation and Sustainability Model (PRISM) framework can help identify important contextual factors such as variability in operations and resources, while the Reach, Effectiveness, Adoption, Implementation, and Maintenance (RE-AIM) framework helps evaluate adoption and success throughout the life of the intervention [48,49]. Both the PRISM and RE-AIM frameworks are well-established and encourage investigators to consider important elements that inform the generalizability of study findings [50]. These frameworks can also encourage investigators to consider the role of both the laboratory and the clinical staff in the implementation, addressing barriers to change and identifying keys to success. The PRISM framework is one example of how the D&I methodology can be used to implement multidisciplinary stewardship interventions (Figure 1). In the case of a laboratory-based diagnostic stewardship intervention, laboratory leaders and staff hold the organizational perspective as those delivering the intervention. Leadership perspectives may include cost, organizational goals, and staffing, while laboratory staff perspectives may include assay training, test ease of use, clinical microbiology education or skill level, and workflow considerations. Microbiologists and clinicians should consider the patient’s perspective, which includes, but is not limited to, how the intervention impacts the patient’s overall care.

Characteristics of intervention recipients include characteristics of clinical leaders, management, and staff whom the stewardship intervention will directly impact. Understanding what these end-users need from the laboratory to carry out their job and provide quality patient care will impact how the diagnostic intervention is implemented, modified, and maintained. Alternatively, providing laboratory-based education to clinicians helps them understand laboratory needs, workflow, and the purpose and importance of the diagnostic intervention [38]. Finally, identifying end-user characteristics early in the intervention development can assist in identifying barriers to success, mainly if the diagnostic intervention aims to impact clinician behavior. 

Patient perspective and characteristics as a recipient include characteristics of individual patients or a patient population that may lead to the modification of an intervention. For example, microbiology result reporting may differ significantly for stem cell transplant patients as opposed to patients without immunosuppression. In addition, external environment elements may significantly impact the success of an implementation and include the regulatory environment, reimbursement, and community resources [48]. Importantly, regulatory requirements or clinical guidelines may impact settings differently and may limit or support changes to microbiology standard operating procedures or antibiotic prescribing practices regardless of the potential impact of a diagnostic test. 

Implementation and sustainability infrastructure may arguably be the most essential component of a successful diagnostic stewardship intervention in the post-analytic period. Pertinent elements of this category include a dedicated team of microbiologists, stewards, and clinician end-users, as well as adopter training and support such as result interpretation guidance, real-time laboratory communication, and cross-training between clinicians and microbiologists [38]. In addition, although dependent on hospital and laboratory resources, stewardship interventions should aim to build relationships and communication with adopters through rounding in the microbiology laboratory and creating open lines of communication between providers and microbiologists. Finally, the success and sustainability of a diagnostic stewardship intervention will depend on the adaptability of protocols and procedures and sharing of best practices with other centers [48].

## 4. Variability in Clinical Impact over Time: Implications on Outcome Evaluations

### 4.1. Learning Curves and Temporal Changes in Effect

Evaluating the effects of a diagnostic test may be variable due to the increasing familiarity of clinicians with the test over time, particularly with complex tests, impacting downstream actions based on test results. In contrast to pharmacology interventions which have a direct exposure and effect on patients, diagnostics impact healthcare provider actions in patient management, and the effects are thus mediated through the healthcare providers and their decision making [51]. Consequently, there is often a learning curve that can improve the performance of the diagnostic intervention over time [52]. This learning curve phenomenon is generally reported in new surgical techniques and is described in blood glucose interventions [53,54,55]. Additionally, non-diagnostic antimicrobial stewardship intervention literature describes this carryover effect from clinical case to clinical case for other behavioral interventions [56,57]. As the initial impact of an intervention may be low during an initial learning period, formal lead or run-in periods should be considered. These are occasionally implemented in clinical trials before assessing differences of the control vs. intervention to reflect the full effect of the intervention more accurately. In contrast, including the run-in period may dilute the average effect of the intervention observed [58,59]. This phenomenon has been observed in the literature with rapid infectious diseases diagnostics, albeit with limited occurrence, as studies generally do not evaluate effect over observable time (e.g., interrupted time series analysis). 

One example of the change in diagnostic effect over time was in a recent study from Kim et al. [60]. They assessed the real-world impact of a sample-to-answer multiplex PCR for respiratory tract infections among children presenting to the hospital emergency department or within their first two days of hospital admission if the patient had a febrile and respiratory presentation. During period I (November 2015 to June 2016), a non-sample-to-answer multiplex respiratory PCR was used while the sample-to-answer multiplex respiratory PCR was used in periods II and III (July 2016 to June 2017 and July 2017 to July 2018, respectively). Outcomes evaluated included the impact on result turnaround time, antibiotic use, and hospital length of stay (LOS). LOS was decreased in period I vs. period III (3.2 days vs. 3.0 days, *p* = 0.004), while no significant difference was observed between period I vs. period II, and period I vs. period II-III. Similarly, intravenous (IV) antibiotic use frequency was decreased in period I vs. period III (51.7% vs. 39.4%, *p* = 0.002) while no significant difference period I vs. period II and period I vs. period II-III. Finally, the duration of combined IV and oral antibiotic use was decreased in period I vs. period III (3.4 days vs. 2.7 days, *p* = 0.0190. However, no significant difference for combined IV and oral antibiotic use was observed in period I vs. period II or period I vs. period II-III. These data speak to both the occurrence of a learning curve and the often importance of designing a run-in period to avoid dilution of the full effect of the diagnostic intervention. 

Another example of this concept occurred in a recent study from Crook et al. [5]. In this study, the investigators evaluated the impact of clinical guidance and of a multiplex respiratory PCR and a multiplex PCR for meningitis/encephalitis on clinical outcomes among febrile or hypothermic infants ≤90 days of age presenting to the emergency department. This study was divided into three periods: period I (January 2011 to December 2014), where no standard local clinical guidelines were established and rapid diagnostics were not implemented, period II (January 2015 to April 2018), where a clinical guideline was implemented, and period III (May 2018 to June 2019), where rapid diagnostic testing was implemented in conjunction with the clinical guideline. Outcomes evaluated included antimicrobial use, ancillary testing, LOS, admission rate, and 30-day mortality. The study included 5317 patients with 2541 in period I, 2082 in period II, and 721 in period III. In an interrupted time series model, the introduction of the guidelines was associated with significant decreases in ancillary testing and lumbar punctures, while the introduction of rapid diagnostic testing was associated with additional decreases in ancillary testing and an increased proportion of infants 29–60 days of age being cared for without antibiotics. Notably, examining the level and slope changes in the time series figures for these outcomes, shows a continued progressively improving outcome over the years during period III. Again, these data lend to the utility of incorporating analyses that account for the learning curve through a run-in period or temporal methods, which can reflect the changing impact over time. However, two other studies involving rapid diagnostics in Gram-positive and Gram-negative bloodstream infections among pediatrics and adults, respectively, observed improvements in changes in therapy by the level but not slope of use over time [61,62]. Thus, these variations related to learning and diagnostic effectiveness improvement over time may be variable across settings.

### 4.2. Changes in Standard of Care or Available Treatments Impact on Diagnostic Effect

Careful consideration of any changes in the standard of care or available treatments should be made with regard to measuring diagnostic effects. For instance, before a drug is introduced for a disease, the diagnostic could potentially lead to changes in infection control or ancillary testing but not changes in individual patient clinical outcomes. However, after introducing a new therapy, the diagnostic test may allow for timely and targeted therapy, possibly leading to improved clinical outcomes for the patient. Similarly, if a new drug is introduced to the market that may improve outcomes in targeted patients compared to the previous standard of care therapy, a diagnostic may allow the rapid identification of patients that would benefit from that therapy. It is important to consider these changes over time when evaluating diagnostic clinical outcome effects.

An example of a new drug introduced to the market for a previously symptomatically managed disease includes new therapies for patients with SARS-CoV-2 [60]. For one of these therapies, in a randomized controlled trial, 2246 patients at high risk of progressing to severe illness were enrolled, and the new medication reflected an 89% reduction in COVID-19-related hospitalization or death from any cause compared to the placebo in patients treated within three days of symptom onset [63]. Before the newer therapies, rapid diagnostic testing of SARS-CoV-2 infection allowed for isolation but did not facilitate RCT-proven therapies to mitigate hospital admissions or death among outpatient diagnosed patients. Following the introduction of these newer therapies, rapid diagnostics for SARS-CoV-2 may allow for rapid, targeted therapy that can lead to improved clinical outcomes for patients in the outpatient setting. 

Similarly, newer improved therapeutics compared to past standards of care therapies may lead to changes in diagnostic effect when introduced. A potential example of this includes newer antimicrobials for multidrug resistant organisms (MDROs) when the diagnostic can guide targeted therapy to patients infected with MDROs. For instance, for carbapenem-resistant Enterobacteriaceae, newer beta-lactam/beta-lactamase inhibitor combinations have been shown to have decreased mortality and adverse effects as compared to aminoglycoside- and colistin-containing regimens employed in such clinical cases [64]. As rapid diagnostics have been shown to decrease time to effective and optimal therapy for extended-spectrum beta-lactamase- and carbapenemase-producing Enterobacteriaceae, the combination of better therapies with more rapid administration has the potential to yield an effect change that could occur over time and should be considered when evaluating the impact of these advanced technologies with newly introduced therapies [65].

## 5. Conclusions

Factors beyond the analytical performance of a diagnostic can impact the effectiveness of a diagnostic test on clinical outcomes. Modifications can be made at the pre-analytical and post-analytical periods to improve the effectiveness of a diagnostic through interdisciplinary collaborations. These include workflow optimization and CDSS for pre-analytical, antimicrobial stewardship, microbiology reports, bespoke interventions based on local context, and implementation science to improve sustainability and success of adoption for the post-analytical phases. Moreover, diagnostic effectiveness can be impacted by changes over time, including learning curves on using the diagnostic and introduction of novel therapies that improve the result’s impact on clinical outcomes. As laboratory and antimicrobial stewardship professionals, we believe diagnostic stewardship is an essential interdisciplinary endeavor to improve diagnostic effectiveness and patient care.

## Figures and Tables

**Figure 1 antibiotics-11-00250-f001:**
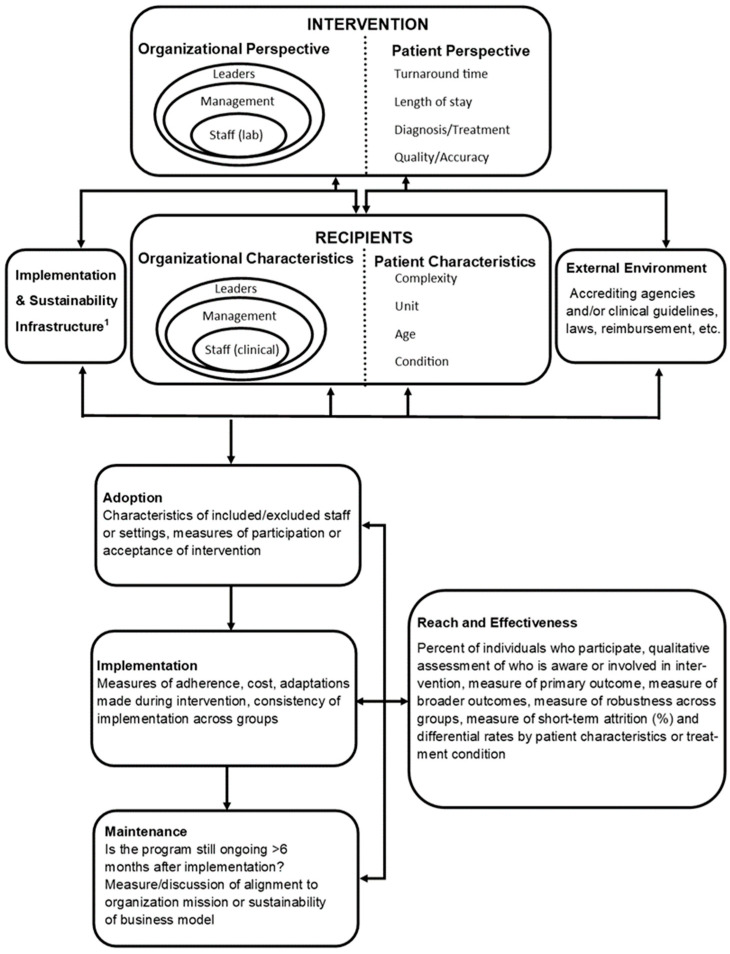
PRISM and RE-AIM frameworks for a post-analytic diagnostic stewardship intervention. ^1^ This may include any personnel that are necessary for a successful implementation. In the case of a diagnostic stewardship intervention this may include trained microbiologists and laboratory staff, a microbiology director or pathologist, an antimicrobial stewardship team, a quality improvement representative, administrators, and information technology specialists. Figure adapted from Feldstein, A.C.; Glasgow, R.E. A practical, robust implementation and sustainability model (PRISM) for integrating research findings into practice. *Jt. Comm. J. Qual. Patient Saf.*
**2008**, *34*, 228-243. Reprinted with permission from ref. 48. 2008. Elsevier.

**Table 1 antibiotics-11-00250-t001:** Potential targets for enhanced implementation of diagnostic interventions and related effect measurement.

Domain	Variations	Implications	Example
Pre-analytical	Optimizing test utilization Use of LIS data & CDSS	Diagnostic stewardship interventions Prevention of duplicate orders Improved knowledge of intended test use Optimized test utilization	Restricting *C. difficile* orders to prevent duplicate or unnecessary testing significantly optimizes testing [3]
Post-analytical	Antimicrobial Stewardship PAF CDSS Templated microbiology comments Local contextual factors (e.g., surgery service) Implementation and dissemination science	Optimization of antimicrobial use Evidence-based guideline use at the point of care Provides microbiological expertise that assists with clinical management Conforms interventions to local needs and abilities Uses scientific methods to improve sustainability and adoption of intervention(s)	Templated comments on *Candida* spp. in urine as normal flora unless high risk was associated with significant decrease in antifungal use [4]
Temporal Changes	Learning curve Changes in standard of care or available treatments	Run-in period before evaluating effects Reassess diagnostic effectiveness with introduction of novel therapies	Performing and plotting interrupted time series analysis of respiratory and CNS testing reflected increasing effectiveness over time [5]

CDSS—clinical decision support system; PAF—prospective audit and feedback.

## Data Availability

Not applicable.

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
