# Peer review of "Diagnostic Stewardship as a Team Sport: Interdisciplinary Perspectives on Improved Implementation of Interventions and Effect Measurement"

_antibiotics, 2022, doi:10.3390/antibiotics11020250_

Round 1

Reviewer 1 Report

Although the review/tutorial is well written, it obviously promotes bioMérieux products. It could be useful if the references to these products have been omitted. The sections promoting the diagnostic products are: a. The lines 93-98 b. The lines 305-342

Author Response

We thank you for your comments. We have removed specific naming of our products as requested by the editor (lines 93-94, 304-309, 322-323). We have maintained the references to inform readers. Of note, we wrote generally to our product based on our knowledge of that literature but also included references to other diagnostic platforms (e.g. Verigene).

Reviewer 2 Report

Interdisciplinary approach to improve implementation of diagnostic stewardship to optimise antimicrobial use is an important area of research. This review informs potential optimization opportunities and the temporal factors that can yield changes in diagnostic effectiveness. The review outcomes have been well written however, there are much room for improvement of the method section, a scientific flaw of this manuscript. 

  1. It is not clear what kind of review it is? Is it a scoping review or a systematic review or a narrative review?
  2. Method section is a completely missing in the manuscript
  3. Where the reviewed studies were come from and how data were  analysed and/or interpreted? 
  4. The PRISM and RE-AIM Frameworks have been used in the figure 1. It is not clear how authors have deduced the figure and how two frameworks have been merged. Is it an established framework or authors have produced/modified by themselves.
  5. Components of framework need more explanations to understand by readers
  6. Review limitation has not been mentioned. 
  7. Though the review has been comprehensively written but some structural changes highlighting key outcomes could improve readership
  8. Methods, results, discussion and conclusion sentences are missing in the abstract. Structuring abstract by improving those aspects could be recommended. 
  9. When, how and who reviewed articles should be clearly described in the paper. 

Author Response

We appreciate the consideration you have given the manuscript. We developed the manuscript as a narrative review, not a systematic literature review (SLR) or similar variation of scientifically reproducible review with targeted features. Therefore, as a narrative review, SLR components (methods, results, etc) would not apply to this article. We have added acknowledgement of the article as a ‘narrative review’ in the abstract and text body on lines 16 and 37. The PRISM and RE-AIM frameworks are previously established and often used together in D&I research. Language to reflect this has been added at lines 236-238, as well as a relevant citation documenting the history of the two frameworks with an example of the merged figure which was adapted for this manuscript.

Reviewer 3 Report

This review is  my opinion is interesting and well written and deserves to be published. 
Would consider to add another table with a good example of diagnostic intervention implementation and related effect measurement (Table 1 demonstration). 

Moreover, it might be interesting to add a section with the minimum laboratory requirements for the pre-analytical phase (based on literature evidence) as the authors point out that there is extreme variability in knowledge and opinion among hospitals and between departments within the same hospital. For example PROS and CONS for rapid molecular diagnostics without rapid respiratory diagnostics.

Author Response

This review is my opinion is interesting and well written and deserves to be published.

Would consider to add another table with a good example of diagnostic intervention implementation and related effect measurement (Table 1 demonstration).

We have expanded Table 1 to include examples of diagnostic intervention implementation and related effect measurement.

Moreover, it might be interesting to add a section with the minimum laboratory requirements for the pre-analytical phase (based on literature evidence) as the authors point out that there is extreme variability in knowledge and opinion among hospitals and between departments within the same hospital. For example PROS and CONS for rapid molecular diagnostics without rapid respiratory diagnostics.

Thank you for this observation. We have added a brief discussion of minimum laboratory specimen requirements for testing to the pre-analytical section of the manuscript (lines 135-153).

Reviewer 4 Report

The manuscript by Kyle D. Hueth et al. gives an overview and general recommendations in the field of clinical microbiology diagnostics, highlighting the importance coordinating local laboratory practices with clinicians and other healthcare providers. The text addresses pre- and post- analytical questions, the need for accurate, timely and actionable results, as well as perspectives on dissemination and translational science. To my opinion, this is a useful review for people working in the field (IVD companies, clinicians, researchers, microbiologists). It is well written and references are all recent or relatively recent and appeared relevant. I am overall positive about this article with minor remarks below, but I would also point out that the authors, who are employees of bioMérieux, advertise a lot about the BioFire FilmArray products commercialized by BioMérieux. Although this is certainly acceptable and backed by referenced studies, I would recommend to cite more competitors, when possible, so as not to give the impression that this review is only a promotion of the company's products.

Suggestion (regarding both a competitor and the future of molecular microbiology lab): As the authors discussed the complexity of existing molecular tests menu either mono-plexed or multiplexed, I will find interesting to develop the promises and current practices of One-For-All molecular microbiology tests based on Next Generation Sequencing. Companies such as Karius (I am not working at Karius) or clinical associated research labs such as the one of Charles Chiu (I am not part of Charles Chiu's team) gives examples of such approaches. Performances and effectiveness of NGS-based actionable diagnostics of meningitis/encephalitis and pulmonary infections can be compared to PCR-based FilmArray ME and RP panels, respectively, which would certainly be of great interest for readers and decision makers.

Minor remarks:

- A definition of rapid diagnostic assay should be given (nature of the test, turnaround time)
- Line 150-151: "positive blood culture result by rapid diagnostic assay": as blood culture can take several days to turn positive, I do not understand this sentence.
- Table 1: extra bullet points can probably be given in the Pre-analytical section, based on what is described in the text.
- Figure 1: this figure is adapted from another publication. I think this is fine, but my feeling is that it adds nothing to the understanding. Text appeared more relevant, at least to my opinion.
- Can the BioFire RP detect bacterial resistance genes? This is not clear to me and can be better explained and discussed.
- Line 162: ref 27 used the FilmArray RP, please mention this information in main text for the sake of transparency.

Author Response

The manuscript by Kyle D. Hueth et al. gives an overview and general recommendations in the field of clinical microbiology diagnostics, highlighting the importance coordinating local laboratory practices with clinicians and other healthcare providers. The text addresses pre- and post- analytical questions, the need for accurate, timely and actionable results, as well as perspectives on dissemination and translational science. To my opinion, this is a useful review for people working in the field (IVD companies, clinicians, researchers, microbiologists). It is well written and references are all recent or relatively recent and appeared relevant. I am overall positive about this article with minor remarks below, but I would also point out that the authors, who are employees of bioMérieux, advertise a lot about the BioFire FilmArray products commercialized by BioMérieux. Although this is certainly acceptable and backed by referenced studies, I would recommend to cite more competitors, when possible, so as not to give the impression that this review is only a promotion of the company's products.

We appreciate the comments and have removed the commercial names (lines 93-94, 304-309, 322-323) as suggested in addition to adding more publications from other multiplex or similar products (lines 97-99, 362-366). We have also removed trade names from the section “Variability in Clinical Impact Over Time: Implications on Outcome evaluations” in addition to adding examples from other molecular products (lines 353-356, 362-366).

Suggestion (regarding both a competitor and the future of molecular microbiology lab): As the authors discussed the complexity of existing molecular tests menu either mono-plexed or multiplexed, I will find interesting to develop the promises and current practices of One-For-All molecular microbiology tests based on Next Generation Sequencing. Companies such as Karius (I am not working at Karius) or clinical associated research labs such as the one of Charles Chiu (I am not part of Charles Chiu's team) gives examples of such approaches. Performances and effectiveness of NGS-based actionable diagnostics of meningitis/encephalitis and pulmonary infections can be compared to PCR-based FilmArray ME and RP panels, respectively, which would certainly be of great interest for readers and decision makers.

Thank you for this comment. As this manuscript focuses on the pre-analytical and post-analytical phases of diagnostic testing and related impact on clinical effectiveness, we have elected not to make any comparisons between these platforms in the analytical phase.

Minor remarks:

- A definition of rapid diagnostic assay should be given (nature of the test, turnaround time)

We have added these defining characteristics on line 90.

- Line 150-151: "positive blood culture result by rapid diagnostic assay": as blood culture can take several days to turn positive, I do not understand this sentence.

This statement has been revised and now reads: “For example, in a pre-post study of adults admitted to the intensive care unit (ICU) with a positive blood culture organism identified by rapid diagnostic assay”

- Table 1: extra bullet points can probably be given in the Pre-analytical section, based on what is described in the text.

Bullets have been added to table 1 to better reflect the narrative in the pre-analytical section. These include the addition of “use of LIS data & CDSS” for variations and “prevention of duplicate orders”, “improved knowledge of intended test use”, and “optimized test utilization” implications.

- Figure 1: this figure is adapted from another publication. I think this is fine, but my feeling is that it adds nothing to the understanding. Text appeared more relevant, at least to my opinion.

We prefer to retain the figure to visually engage some readers on details of the D&I framework. We believe emphasis on D&I for improving diagnostic effectiveness is critical (e.g. EBM taking 17 years to reach the bedside) and sparse in the diagnostic literature.

- Can the BioFire RP detect bacterial resistance genes? This is not clear to me and can be better explained and discussed.

- Line 162: ref 27 used the FilmArray RP, please mention this information in main text for the sake of transparency.

We have removed specific naming of our products as requested by the editor (lines 93-94, 304-309, 322-323). Given this the editor’s consideration, we have also elected to refrain from describing specific diagnostic test characteristics (e.g. bacterial resistance genes).

Round 2

Reviewer 1 Report

The manuscript has been improved considerably.

Reviewer 2 Report

This is a much improved version.